# *Pouteria macrophylla* Fruit Extract Microemulsion for Cutaneous Depigmentation: Evaluation Using a 3D Pigmented Skin Model

**DOI:** 10.3390/molecules27185982

**Published:** 2022-09-14

**Authors:** Ana Clara N. Brathwaite, Thuany Alencar-Silva, Larissa A. C. Carvalho, Maryana S. F. Branquinho, Ricardo Ferreira-Nunes, Marcilio Cunha-Filho, Guilherme M. Gelfuso, Silvya S. Maria-Engler, Juliana Lott Carvalho, Joyce K. R. Silva, Tais Gratieri

**Affiliations:** 1Laboratory of Food, Drugs, and Cosmetics (LTMAC), Health Sciences Faculty, University of Brasilia, Brasilia 70910-900, DF, Brazil; 2Laboratory of Genomic Sciences and Biotechnology, Catholic University of Brasilia, Brasília 71966-700, DF, Brazil; 3Department of Clinical and Toxicological Analysis, School of Pharmaceutical Sciences, University of São Paulo, São Paulo 05508-000, SP, Brazil; 4Interdisciplinary Laboratory of Biosciences, Medicine Faculty, University of Brasilia, Brasília 70910-900, DF, Brazil; 5Laboratory of Biotechnology of Enzymes and Biotransformation (LABEB), Biological Sciences Institute, Federal University of Pará, Belém 66075-110, PA, Brazil

**Keywords:** cutite extract, gallic acid, hyperpigmentation, microemulsion

## Abstract

Here, we verify the depigmenting action of Pouteria macrophylla fruit extract (EXT), incorporate it into a safe topical microemulsion and assess its effectiveness in a 3D pigmented skin model. Melanocytes-B16F10- were used to assess the EXT effects on cell viability, melanin synthesis, and melanin synthesis-related gene transcription factor expression, which demonstrated a 32% and 50% reduction of intra and extracellular melanin content, respectively. The developed microemulsion was composed of Cremophor EL^®^/Span 80 4:1 (*w*/*w*), ethyl oleate, and pH 4.5 HEPES buffer and had an average droplet size of 40 nm (PdI 0.40 ± 0.07). Skin irritation test with reconstituted epidermis (Skin Ethic RHE^TM^) showed that the formulation is non-irritating. Tyrosinase inhibition was maintained after skin permeation in vitro, in which microemulsion showed twice the inhibition of the conventional emulsion (20.7 ± 2.2% and 10.7 ± 2.4%, respectively). The depigmenting effect of the microemulsion was finally confirmed in a 3D culture model of pigmented skin, in which histological analysis showed a more pronounced effect than a commercial depigmenting formulation. In conclusion, the developed microemulsion is a promising safe formulation for the administration of cutite fruit extract, which showed remarkable depigmenting potential compared to a commercial formulation.

## 1. Introduction

*Pouteria macrophylla* (Lam.) Eyma belongs to the Sapotaceae family and is found in the Brazilian Amazon region and other South American countries, including Bolivia, Peru, French Guiana, Suriname, Colombia, and Venezuela. Its fruits possess common names such as cutite, jarana, abiurana-cutite, taturubá, and abiu-cutite, among others. The fruit pulp can be consumed fresh or in the manufacture of ice cream and sweets, while the fruit peel is popularly used to treat dysentery [1]. Chemically, the cutite fruit is characterized by the presence of phenolic compounds, such as gallic acid (GA), p-coumaric, vanillic, ferulic, 3,4-dihydroxybenzoic, synaptic, caffeic, and flavonoids such as quercetin and catechins [2,3,4]. Furthermore, the presence of GA as the main compound and biomarker, followed by other phenolic compounds, promotes its antioxidant action [2,5]. Nonetheless, more than the antioxidant activity, these compounds present a cosmetic interest due to their anti-melanogenic action by inhibiting tyrosinase and suppressing other genes related to pigment formation [6]. Indeed, the tyrosinase inhibition potential of GA has already been described [7,8].

Hyperchromias are anomalies that occur in the skin due to the accumulation of melanin or an increase in melanocytes number [9,10]. Exposure to ultraviolet (UV) radiation is the main factor related to the appearance of hyperchromias, triggering several reactions that increase the transfer of melanin to keratinocytes [11]. The main form of hyperchromia treatment is the use of depigmenting actives. These substances can act in several ways, mainly through the suppression or inhibition of tyrosinase synthesis or genes related to melanogenesis, such as the transcription factor associated with microphthalmia (MITF) [12,13]. Tyrosine, an essential amino acid, is the initial element of melanin biosynthesis through the action of tyrosinase, giving rise to eumelanin and pheomelanin. In the presence of molecular oxygen, tyrosinase oxidizes and transforms tyrosine into DOPA and this into dopaquinone [8,14].

The use of depigmenting actives with tyrosinase inhibitory mechanisms is increasingly common in the cosmetic industry. However, prolonged use can trigger several adverse effects. These compounds generally have high cytotoxicity and can generate mutations in melanocytes. Hydroquinone, considered the gold standard for treating hyperchromias, can cause contact dermatitis, skin irritation, and exogenous ochronosis [15]. In addition, the topical application effectiveness of such compounds can be affected by a low capacity to penetrate the skin and low stability in the formulations. Such limitations encourage the use of novel naturally occurring depigmenting actives [8,16]. The antimelanogenic action of GA sets a precedent for the research of the cutite fruit extract as an active pharmaceutical ingredient. Nevertheless, more importantly, the formulation must maintain extracts’ composition stably and have an appealing sensorial feeling, which is indispensable in cosmetics.

In this sense, microemulsion appears as an ideal formulation. Microemulsions are liquid, transparent, thermodynamically stable nanosystems of water and oil stabilized by a surfactant and co-surfactants [17]. They have been demonstrated to be a suitable carrier for plant extracts, preserving the stability and enhancing cutaneous delivery [18].

Hence, the present work initially aimed to verify for the first time the cutite extract depigmenting action by cell assays using melanocytes. Next, a stable cutite extract-loaded microemulsion was obtained and evaluated by in vitro release and permeation assays. Finally, the proposed formulation was challenged by a 3D culture model of pigmented skin envisaging a safe and effective cutaneous administration for cosmetic purposes.

## 2. Results and Discussion

### 2.1. Cutite Fruit (Pouteria macrophylla) Lam. Emya Extract Characterization

The EXT consists of several phenolic actives, with GA as a major phenolic component. The GA content in the extract was calculated to be 13.4 mg/g DW, i.e., 1.34%. These values were higher than those previously reported in the literature for fresh weight (4.72 mg/g FW) and very similar in relation to dry matter (12.5 mg/g DW) [2]. Another major component was quercetin (0.044 mg/g DW), which was in accordance with the literature [5].

### 2.2. Melanocyte Cell Culture and In Vitro Pigmentation Studies

#### 2.2.1. Cell Viability Assay

The MTT assay was performed to verify the viability of B16F10 cells treated with DMSO, EXT, and GA, aiming to select safe concentrations for further depigmenting assays. As a result, the concentrations of the standard tested are within the typical values for this type of experiment [7].

Compared to the untreated controls, which were 100% viable and used to normalize the acquired data, the concentration of 0.05 μg/mL of EXT (Figure 1A), containing 2.968 x 10–6μM of GA and GA 100 μM (Figure 1B) was selected to continue the assays due to its lack of toxicity compared to the control.

#### 2.2.2. Depigmenting Action of Cutite Extract

A relative quantification assay for melanin was performed to assess whether EXT and GA promoted a decrease in melanin production by melanocytes. Intracellular melanin content was normalized according to IBMX-stimulated cells, in which the melanin content was considered 100% (positive control). Unstimulated cells were used as negative control (57.29% ± 0.14%).

Compared to stimulated cells (100%), GA (55.3 ± 0.05%) and EXT (68.6 ± 0.01%) promoted a significant decrease (*p* < 0.0001) in intracellular melanin content (Figure 2A). The GA reduced the production of intracellular melanin by 44.69%. This result follows the literature, demonstrating the effectiveness of using GA to treat hyperchromias [7,19]. Despite the low concentration of EXT, based on its cytotoxicity, there was also a significant reduction of 31.42% in the production of intracellular melanin. This result demonstrates the effectiveness of the depigmenting action of EXT even at low, non-toxic concentrations.

The treatment of B16F10 with GA (48.37 ± 0.008%) and EXT (49.75 ± 0.003%) also promoted a decrease in the melanin secretion in the culture supernatant (Figure 2B). The GA and the EXT reduced the extracellular melanin by 51.63% and 50.25%, respectively. This result reinforces the efficacy of the depigmenting action of GA and EXT. In addition, the extract has other assets, such as ferulic acid and catechin, which also have depigmenting activity and may be acting in synergy with GA, explaining its effective depigmenting action even at lower concentrations when compared to GA alone [20,21].

#### 2.2.3. Melanogenesis Gene Suppression

The melanin production pathway involves several essential regulatory genes. MITF is the principal regulator of melanogenesis. Its function is to regulate the gene expression of tyrosinase, dopachrome tautomerase (DCT), and TRP-1 enzymes. Furthermore, MITF regulates melanocyte differentiation, pigmentation, proliferation, and viability [22,23].

The tyrosinase and MITF genes involved in melanogenesis were evaluated to confirm the depigmentation potential of GA and EXT at previously established concentrations.

The exposure of B16F10 to treatment with GA (0.009 ± 0.005%) and EXT (0.900 ± 0.178) significantly suppressed the relative gene expression of MITF (Figure 3A). According to Kumar et al. [19], GA not only inhibits the expression of MITF but also suppresses its nuclear translocation.

Tyrosinase gene expression was suppressed after treatment with GA (2.457 ± 0.413) and EXT (2.09 ± 0.31), as shown in Figure 3B. Such a result was expected due to the regulation carried out by MITF in tyrosinase transcription [22]. Therefore, it is possible to hypothesize that the melanin content reduction observed in the cells involved the modulation of the mRNA expression of essential genes concerned in melanin synthesis.

Our observations were made in the context of IBMX-induced melanin synthesis. Importantly, IBMX induction of melanin synthesis by melanocytes replicates relevant physiological mechanisms of melanin synthesis induction. IBMX treatment induces an increase in cAMP levels that activates PKA and leads to the expression of MITF and TYR [11,23]. UV and α-MSH stimulation share such a signaling pathway. Furthermore, even the parallel melanin synthesis pathway of cKit stimulation, a relevant mechanism by which keratinocytes and other cell types stimulate melanin production, concatenates in the induction of MITF and TYR expression. Therefore, the fact that both EXT and GA inhibit the mRNA expression of those components helps to explain the anti-melanin effect observed both at the intracellular and extracellular levels and allows one to suggest that those compounds will possibly be effective against pigmentation disorders with different etiologies, such as UV exposure and melasma, citing a few.

### 2.3. Obtention and Characterization of Microemulsion

Cremophor^®^ and Span 80^®^ are non-ionic surfactants with low toxicity compared to ionic surfactants, therefore they were selected to constitute the microemulsion [24]. Additionally, ethyloleate is reported to cause negligible cutaneous irritation [18]. The pseudoternary diagram is shown in Figure 4.

A stable microemulsion was selected from the pseudoternary diagram based on formulation appearance, consistency, and characteristic maintenance. The selected composition was 50% pH 4.5 HEPES buffer, 40% Cremophor^®^ + Span 80^®^ (4:1 *w*/*w*) and 10% ethyl oleate.

A 37.3 mg/g EXT concentration corresponding to 0.05% GA (based on previous in vitro B10F10 assays) was incorporated into the selected microemulsion. The formulation exhibited a yellow translucid aspect, surfactant smell, turbidity absence with homogeneity. The microemulsion presented a droplet size of approximately 48.7 ± 4.7 nm (PDI 0.35 ± 0.09). The zeta potential was negative (−25 ± 1.42 mV) at the formulation pH (4.0 ± 0.1). Negative zeta potential was expected from the use of Span^®^ 80 as a surfactant [25]. The TEM image in Figure 5 shows the microemulsion’s morphology, which agrees with the dynamic light scattering measurements.

### 2.4. Short-Term Stability Study

The zeta potential of the microemulsion varied from −25.0 ± 1.4 to −6.3 ± 0.7, and the droplet size and PDI of the microemulsion remained stable for 30 days (Figure 6). Also, there was no visual modification and instability signs, such as turbidity, sedimentation, or coalescence [26]. The microemulsion, therefore, was considered physically stable over 30 days. After storage conditions, the pH of the microemulsion oscillated from 4.0 ± 0.1 to 3.7 ± 0.1 within the stability range of GA [27]. Longer stability studies must be performed. If the tendency of pH drop is confirmed, preservatives must be added to the formulation in future studies.

The microemulsion maintained its GA content for at least 30 days even without adding an antioxidant to the formulation. On the other hand, the conventional emulsion showed a significant loss in GA content, with a reduction of approximately 50% (*p* < 0.05). GA is a hydrophilic molecule remaining in the aqueous phase of the emulsions and therefore has more contact with oxygen in the environment, increasing the oxidation processes of the molecule and its consequent degradation [28]. In the case of the microemulsion, even with the drug tending to also be located in the external phase, the higher concentration of surfactant and co-surfactant in the formulation may allow the drug to better interact with these molecules, locating closer to the surface of the nanodroplets, which preserves its chemical stability. Similar results have been described in the literature [18].

### 2.5. Skin Irritation Test

The EpiSkin™ RHE model assesses the irritant potential of chemical agents through the initial events of the inflammatory cascade by measuring cell viability. Compounds are considered non-irritating when they have cell viability greater than 50% of the negative control cell viability. The acceptable standard deviation for the test should be <18% [29].

The non-irritating negative control exhibited 100 ± 5% of cell viability, while the irritating positive control showed a cell viability of 6.75 ± 1.00%, well below the reference value (≤50%) for irritating compounds. Conventional emulsion presented 54.22 ± 1.00% and microemulsion 75.54 ± 1.00% negative control cell viability (Figure 7). According to the guidelines presented in TG 439 [29], both formulations are classified as non-irritating and guarantee safety for their topical application. Therefore, despite the high concentration of surfactants in microemulsion (40%), it seems that the choice of non-ionic surfactants, which is recognized by low irritating potential [24], contributed to the formulation’s safety.

### 2.6. In Vitro Release Study

GA release profiles from conventional emulsion and microemulsion were obtained for 12 h. An aqueous solution was also used as a control.

In the first hour, 58.5 ± 5.0% GA was released from the aqueous solution. In contrast, 16.6 ± 2.4% and 9.6 ±1.9% were released from conventional emulsion and microemulsion, respectively (Figure 8). In the third hour of the experiment, 100% GA was released in an aqueous solution to the receiving medium. The maximum GA release was achieved after 8 h from the conventional emulsion (32.3 ± 1.83%) and after eleven hours from the microemulsion (27.8 ± 2.07%), confirming the expected controlled release from the latter.

### 2.7. In Vitro Skin Permeation

This study verified the EXT permeation through the skin, measuring the tyrosinase inhibition of the skin extracts and receptor solution after 6 h of in vitro treatment with the microemulsion compared to the conventional emulsion.

EXT that permeated through the skin into the receptor solution from microemulsion and conventional emulsion caused tyrosinase inhibition, respectively, by 3.33 ± 1.04% and 2.73 ±1.01%, compared to the positive control (Figure 9A). The reduced effect at tyrosinase inhibition suggests a low availability of EXT in the receptor solution, demonstrating a low EXT permeation through the skin. This finding is a positive result considering the significant distribution of the EXT active compounds in the more superficial skin layers, ultimately avoiding a systemic distribution and possible associated adverse effects.

Figure 9B,C show tyrosinase inhibition by the EXT that penetrated respectively the stratum corneum stratum and remaining skin, which comprises the viable epidermis and part of the dermis. Notably, the microemulsion provided doubled inhibition than the conventional emulsion (20.70 ± 2.18 and 10.69 ± 2.40%, respectively) at the remaining skin, which is the target site for the depigmenting action, as melanocytes are present in the basal layer of the epidermis.

### 2.8. Formulation Efficacy on 3D Pigmented Reconstructed Human Skin Model

A pigmented human skin model was obtained from reconstructed human skin induced for more melanin production. Differentiated epidermal layers and melanin spots (indicated by the arrows) can be observed in the untreated skin (Figure 10A), confirming that the model was functional. There is currently no commercial product for a depigmenting action containing GA. So, for comparison, a formulation containing kojic acid, with is already a well-known depigmenting active was selected. This particular kojic acid serum was chosen as a positive control because it is a commercial formulation from a notorious brand with a positive perception from users based on comments on the seller’s websites. Indeed, lightening of the melanin spots and a decrease in the number of melanin spots, in general, could be observed in the skin treated with the commercial product (Figure 10B), as indicated by the arrows. Notably, in the skin treated with the microemulsion containing the extract (Figure 10C), the melanin spots were also lightened. However, there was a more accentuated decrease in the number of melanin spots compared to the other treatments (non-treated control and skin treated with commercial formulation). This result clearly reinforces the greater effectiveness of the nanostructured system containing EXT compared to a commercial depigmenting formulation, evidencing its great potential for cosmetic use.

## 3. Materials and Methods

### 3.1. Chemical and Reagents

GA (>99%), HEPES (2-[4-(2-hydroxyethyl)-piperazin-1-yl]-ethanesulfonic acid), Span^®^ 80, Cremophor^®^ ELP, mineral oil, Polawax^®^, butylated hydroxytoluene (BHT), L-tyrosine, and tyrosinase were purchased from Sigma-Aldrich (Steinheim, Germany). Ethyl oleate, 98% formic acid for LC-MS, and ammonium chloride were purchased from Merck (Darmstadt, Germany). Methanol and ethanol of chromatographic grade were acquired from J.T. Baker (Phillipsburg, NJ, USA). The constituents of the emulsions, methylparaben, propylparaben, and propylene glycol, were obtained from Dinâmica Química Contemporânea Ltd.a. (São Paulo, Brazil). Dow Corning (DC) 556 and 2501 were purchased from Dow Corning Corporation (Midland, TX, USA). EDTA-disodium was obteined from Vetec Química Fina Ltd.a. (Rio de Janeiro, Brazil). Pre-cleaned filters, 25 mm in diameter and with 0.45 μm hydrophilic pores, were purchased from Analitica (São Paulo, Brazil). All analyzes were performed with ultrapure water (Millipore, Illkirch-Graffenstaden, France). MTT (3-(4,5dimethyl-2-thiazolyl-2) -2,5-diphenyl-tetrazolium bromide) and isoproterenol were purchased from Sigma Chemicals (St. Louis, MO, USA). Dulbecco’s modified Eagle medium (DMEM), fetal bovine serum (FBS), and Trizol were purchased from Gibco Life Technologies (Indianapolis, IN, USA). Penicillin/streptomycin was obtained by Invitrogen (Grand Island, NY, USA). RAFT: KGM-Gold Bullet Kit medium and 254 medium were purchased from Lonza (Walkersville, MD, USA). Type-I collagen gel (354236-I) was obtained from Corning (Tewsbury, MA, USA). Ampicillin sodium salt and streptomycin were acquired from Gibco Life Technologies (Grand Carlsbad, CA, USA). B16F10 immortalized murine melanoma cells were donated by the Molecular Pharmacology Laboratory (FarMol) of the University of Brasilia (Brasília, Brazil) but were originally obtained from the American Type Culture Collection (ATCC). The reconstructed human epidermis (RHE) test method kit was kindly donated by EpiSkin™ (Rio de Janeiro, Brazil). Scotch book tapes no. 845 (3 M, St. Paul, MN, USA) were used to perform the tape stripping technique. Skin from porcine ears was obtained from Sabugy Agroindustria e Comercio de Alimentos (Brasilia, Brazil). For the 3D pigmented skin model, human melanocytes, keratinocytes, and fibroblasts were isolated from donated foreskin samples from the University of São Paulo Hospital (São Paulo, Brazil). The cells were isolated, as previously described by Pennacchi et al. (PENNACHI et al., 2015), under the approval of the local Ethics Committee (HU CEP Case # 943/09, SISNEP CAAE 0062.0.198.000-9). Commercial formulation (Kojic Acid Serum, Botik, SP, Brazil) used as a control in a 3D pigmented skin model was purchased in a regular store in Brasilia.

### 3.2. Cutite Fruit (Pouteria Macrophylla) Lam. Emya Extract Obtention and Characterization

The fruits were collected, and the aerial parts and fruits were identified by comparison with authentic vouchers of *Pouteria macrophylla* (Lam.) Eyma existing in the Herbarium of the Museum Emílio Goeldi (MG239766), city of Belém, state of Pará, Brazil. Fruits were freeze-dried for 48 h and then extracted by percolation with ethanol assisted by ultrasound for 10 min, as previously described [2]. After percolation, the solvent was evaporated using a rotating evaporator. The process yield was 42.98%, and the dried extract was named EXT. GA content in the EXT was confirmed by LC-MS, according to the method described in Section 2.2.

### 3.3. LC-MS Analyses

The determination of GA, the biomarker of the cutite fruit ethanolic extract, was performed by liquid chromatography–mass spectrometry (LC-MS) using an equipment model 2020 from Shimadzu (Kyoto, Japan), connected to a Genius NM32LA nitrogen gas generator model (Peak Scientific, United Kingdom) and mechanical vacuum pump (Edwards, Burgess Hill, England), with dual-spray ionization source, acting on negative ionization mode by electrospray (ESI-), coupled with liquid chromatography system.

A C18 reverse-phase column (4.6 mm × 150 mm; 5 μm Shim-pack, Shimadzu) was used as the stationary phase. The mobile phase was composed of 0.1% formic acid in water (A) and 0.1% formic acid in methanol (B) applied by gradient with a flow rate of 0.4 mL/min according to the schedule: A/B = 90/10 (0.01 min–1.5 min), 80/20 (1.6 min–3.5 min), 70/30 (3.6 min–5.5 min), 60/40 (5.6 min–7.5 min), 10/90 (7.6 min–10 min), and 90/10 (10.1 min–15 min). The oven temperature was maintained at 35 °C, and the injection volume of each sample was 2 μL. The limits of detection and quantification were 0.1 μg/mL and 0.3 μg/mL.

### 3.4. Melanocyte Cell Culture and In Vitro Pigmentation Studies

The analysis of the cellular toxicity, depigmentation effect, and suppression of genes involved in melanogenesis by GA and EXT was performed in immortalized melanocyte cultures from murine melanoma (B16F10). The cells were cultured using DMEM supplemented with 10% (*v*/*v*) FBS and 1% (*v*/*v*) of penicillin/streptomycin solution (1000 U/mL).

#### 3.4.1. Cell Viability Assay

EXT and standard analytical concentrations were tested to select the highest non-toxic concentrations for further experiments. B16F10 cells were seeded in 96-well plates at a density of 1 × 10^4^ cells per well and incubated for 48 h. Then, cells were treated with GA and EXT. The concentrations of the EXT tested were 0.001, 0.01, 0.05, 0.1, 1, and 10 μg/mL (*m*/*v*) using DMSO as a solvent and GA was used at 10, 25, 50, 100, 200, and 400 μM. Cells incubated with vehicle only were considered as untreated controls. After 48 h, cells were washed with PBS, fresh DMEM was supplemented with FBS, and 5 mg/mL MTT was added, followed by plate incubation for 4 h in the dark. Once the MTT solution was carefully removed, DMSO was added to dissolve the formed formazan crystals. A spectrophotometer of microplates (BioTek, PowerWave, VT, USA) was used to read the microplate at 570 nm. After deducting the optical density obtained from the blank (DMSO), the analysis was carried out. Data were normalized to the untreated controls, considered 100% viability, representing the mean ± SD of three independent experiments.

#### 3.4.2. Determination of Intracellular and Extracellular Melanin Content

Melanin production was assessed regarding intracellular melanin production and secretion in the cell culture supernatant, as previously described by our group [17]. Briefly, 3 × 10^5^ cells were plated in 6-well plates and incubated with 0.1 mM 3-isobutyl-1-methylxanthine (IBMX) to induce melanogenesis and treated with GA or EXT, for 48 h. Extracellular melanin content was evaluated by collecting 100 μL of cell culture supernatant from each well. The reading was performed in a microplate reader (Bio-Tec PowerWave, HT, USA) at 405 nm, and the analysis was performed after deducting the optical density obtained in the blank (DMEM). The intracellular melanin content was evaluated from cells harvested, lysed, and mixed with 200 μL of NaOH (1 M) for 16 h at room temperature (20 –25 °C). The absorbance was measured at 405 nm, and the analysis was performed after deducting the optical density obtained in the blank (NaOH). The melanin content of IBMX-stimulated cells was considered 100% and used to normalize melanin production in the experimental groups. Untreated samples received vehicle only and were used as the negative control.

#### 3.4.3. Quantitative Reverse Transcription-Polymerase Chain Reaction (qRT-PCR)

The isolation of total RNA from the cells treated with IBMX, or IBMX and GA or EXT for 48 h was performed using the reagent trizol and following the manufacturer’s instructions to analyze MITF and TYR mRNA expression. First, the RNA concentration was determined by reading the absorbance at 260/280 nm on Nanodrop equipment (Thermo Fisher Sientific, Waltham, MA, USA). Next, the RNA samples were reverse-transcribed using the High-Capacity cDNA Reverse Transcription Kit, following the manufacturer’s instructions. Finally, the mRNA levels were determined by qPCR using the SYBRTM Green Master Mix (Thermo Fisher Sientific, Waltham, MA, USA) and the following primers: GAPDH F - ACATCGCTCAGACACCATG, GAPDH R - TGTAGTTGAGGTCAATGAAGGG; MITF F: AGGACCTTGAAAACCGACAG, MITF R - GTGGATGGGATAAGGGAAAG; TYR F: AGCCTGTGCCTCCTCTAA, TYR R: AGGAACCTCTGCCTGAAA. Amplifications were performed by StepOne Plus equipment (Thermo Fisher Sientific, Waltham, MA, USA), and the data were analyzed by StepOne Software v2.3 [30] using the 2-ddct method.

### 3.5. Preparation and Characterization of Formulations

#### 3.5.1. Microemulsion

The microemulsion was obtained from a pseudoternary phase diagram as previously described [18]. Briefly, HEPES buffer, pH 4.5, was used as the aqueous phase and the surfactants Cremophor^®^ and Span^®^ 80 were mixed in a ratio of 4:1 (*w*/*w*) under vigorous stirring. The surfactants were then dissolved in oil phase (ethyl oleate) in progressive ratios of 9:1, 8:1, 7:1, 6:1, 5:1, 4:1, 3:1, 2:1, and 1:1 (*w*/*w*) at room temperature. The systems obtained from each mixture rested for 5 min and were then classified according to their physical aspect as “phase separation”, “milky”, “cloudy”, “flocculation”, or “possible microemulsion”.

Among the region that forms microemulsion, the composition with the highest viscosity, a higher proportion of pH 4.5 HEPES buffer, and a lower concentration of surfactants was selected to continue the study. Subsequently, 37.3 mg/g of EXT, corresponding to 0.05% of GA content, was added to the chosen microemulsion and mixed with a glass stick for incorporation. The EXT was added at temperatures lower than 40 °C.

The microemulsion was characterized by means of droplet size, polydispersity index (PDI), and zeta potential using a “Zetasizer Nano ZS” equipment (Malvern, Worcestershire, UK) and pH using DM-22 equipment (Digimed, São Paulo, Brazil). Morphological analysis was also performed using diluted samples (1:1000) analyzed by transmission electron microscopy (TEM; JEM 1011 Transmission Electron Microscope, JEOL, Tokyo, Japan–100 kV). The images were captured with a GATAN BioScan camera (model 820, Pleasanton, CA, USA) using the Digital Micrograph 3.6.5 software (Pleasanton, CA, USA).

#### 3.5.2. Emulsion-Gel

An emulsion gel was obtained to serve as a conventional formulation control by preparation of two heated phases separately. Phase 1 comprised 0.4% Aristoflex^®^ AVC, 0.1% EDTA-disodium, 5% propylene glycol humectant, and 94.5% pH 4.5 HEPES buffer. Phase 2 was composed by 14% Polawax^®^, 1% liquid vaseline, 2% DC 556, 1% DC 2501, 0.1% BHT and 0.5% methylparaben and propylparaben. After mild heating (50 °C), phase 1 was poured into phase 2 under vigorous stirring to form the emulsion. After reaching room temperature, the emulsion was homogenized and centrifuged at 4000 rpm for 10 min to assure there would be no phase separation. The pH of the formulation was 4.0 ± 0.3. For comparison purposes, the same amount of EXT was incorporated by geometric dilution technique into the conventional emulsion (37.3 mg/g).

### 3.6. Short-Term Stability Study

Stability studies were carried out for 30 days with formulation samples stored in hermetically sealed Eppendorf at room temperature (20–25 °C) (*n* = 3 for each formulation). At 0, 1, 7, 15, and 30 days, samples were analyzed for droplet size, PDI, and zeta potential for the microemulsion only, and pH and GA content for both the microemulsion and the conventional emulsion formulations.

### 3.7. Skin Irritation Test

The irritative potential of the microemulsion and the conventional emulsion was evaluated using reconstructed human skin (RHE, SkinEthic RHETM) provided by EpiSkin™ according to OECD guidelines described in test n. 439 [29]. For the skin irritation tests, formulations were prepared and stored at room temperature for one week prior to the experiments. Briefly, 16 ± 0.5 μL of microemulsion or conventional emulsion were added over the RHE. The same amount of sodium dodecyl sulfate 5% (*w*/*v*) was used as a positive control, and saline solution at 0.9% (*w*/*v*) as a negative control. The samples remained in contact with the RHE for 42 min in an oven at 5% CO2, 37 °C, and 95% humidity. After this period, each well was washed with 25 jets of 1 mL of saline solution, and the plate was stored for 42 h under the same conditions. Finally, the tissues were transferred to another 24-well plate containing 300 μL/well MTT bromide solution at 1 mg/mL (1:5 dilution at the time of use) and incubated for 2 h protected from light in an oven at 5% CO_2_, 37 °C, and 95% humidity. Subsequently, 750 μL of isopropanol was added under and 750 μL over the tissues for extraction. Then, the reading was performed by colorimetry in a microplate reader (Bio-Tec PowerWave, Winooski, VT, USA) at 570 nm. The study was carried out in triplicate, and the result was based on the optical density obtained from each sample minus the optical density of the blank (isopropanol).

### 3.8. In Vitro Release Study

The release of GA from the EXT incorporated into the formulations was determined in vitro throughout 12 h using modified Franz-type diffusion cells (diffusion area = 1.3 cm^2^) mounted with hydrophilic cellulose membranes (12,000–14,000 MWCO) separating donor and receptor compartments [31]. The donor compartment was filled with 500 mg of the microemulsion, conventional emulsion, or GA aqueous solution (control). The system temperature was kept at 30 °C by a water bath. All experiments followed “sink conditions.” Samples from the receptor were collected hourly.

### 3.9. In Vitro Skin Permeation

In vitro cutaneous permeation studies were performed trough porcine skin using modified Franz-type diffusion cells (diffusion area = 1.3 cm^2^) for 6 h. The donor compartments were filled with 500 mg of microemulsion or conventional emulsion. The system temperature was kept at 30 °C by a water bath. Afterward, the buffer samples were collected and the tyrosinase bioassay method described in Section 3.9.1 was followed for the analysis of the tyrosinase inhibition of the extract. Next, tape striping was performed to remove the stratum corneum. Finally, the remaining skin was cut into small pieces and placed in a glass tube with water and ethanol (1:1 v/v) under stirring at 500 rpm for 16 h for actives extraction. Ultimately, the samples were filtered (0.22 μm), and the tyrosinase bioassay was performed.

#### 3.9.1. Tyrosinase Bioassay

The tyrosinase inhibition was determined by modifying the dopachrome method [32] using an L-tyrosine substrate. In a cuvette, 400 μL of each sample obtained from the in vitro skin permeation assay was placed and added to a tyrosinase solution at 0.1 mg/mL in 800 μL phosphate buffer (0.1 M, pH 6.8). Then, 400 μL of the substrate (L-tyrosine at 0.5 mg/mL) was added and incubated for 30 min at 37 °C. After 30 min of reaction, the absorbance was read at 492 nm, and the inhibition percentage was calculated in relation to the control. Phosphate buffer and kojic acid were tested under the same protocol to be used as negative and positive controls, respectively.

### 3.10. Formulation Activity on 3D Pigmented Reconstructed Human Skin Model

The 3D pigmented reconstructed human skin model was first obtained by our group as previously reported. Basically, normal human melanocytes, keratinocytes, and fibroblasts were cultivated in specific growth medium for each cell type and maintained in an incubator at 37 °C containing 7.5% CO_2_ for keratinocytes and 5% CO_2_ for melanocytes and fibroblasts. Then, pigmented reconstructed human skin was prepared in two steps. First, the dermal compartment was prepared using type-I collagen gel (354236-I, Corning, Tewsbury, MA, USA) and fibroblasts (1.5 × 105/equivalent). After polymerization, 2.5 × 105 human keratinocytes and 1.7 × 105 human melanocytes were seeded in RAFT: KGM-Gold Bullet Kit medium (1:1) on top of each lattice, and the skins were kept submerged in the culture medium for 24 h. The construct was transferred and maintained at an air-liquid interface for 10 days in a 5% CO2 incubator to allow complete keratinocytes stratification and differentiation. The medium was supplemented with tyrosine (0.25 mM) and NH4Cl (5 mM) and was replaced every 2 days to induce pigmentation. The skins were treated with 10 μL of the commercial product (kojic acid serum) or 10 μL of the microemulsion containing the EXT for 12 h. After this, skins were fixed in 10% buffered formalin at 4 °C for 12 h, followed by dehydration in solutions containing increasing concentrations of alcohol and xylene for paraffin inclusion. Paraffin sections (5 μm) were stained with Fontana-Masson. All images were obtained by optical microscopy (Nikon Eclipse- 20× and 40×). The experiment was performed in three independent replicates.

### 3.11. Statistical Analyses

The pseudoternary phase diagram was constructed with Origin 8 (OriginLab Corporation, Northampton, MA, USA). Statistical analyses were conducted using Prism 6 (GraphPad Software Inc., San Diego, CA, USA), in which the level of significance was set at *p* < 0.05, *p* < 0.01, *p* < 0.001, and *p* < 0.0001. Differences between datasets were verified by normality tests via one-way analysis of variance, followed by Tukey’s multiple comparison test.

## 4. Conclusions

The present study demonstrated the depigmenting potential of cutite fruit extract for the first time. Melanocyte cultures showed decreased intracellular melanin production and secretion, along with suppression of melanogenic related genes MITF and TYR. Furthermore, when incorporated in a microemulsion formulation, the extract was slowly released and provided a targeted delivery to the skin layer below the stratum corneum, which is the target site for hyperchromia treatment. Furthermore, the developed microemulsion was shown to be non-irritating to the skin, constituting a safe formulation for topical cutaneous application. Finally, all the promising findings regarding the extract and the formulation potential was confirmed in 3D pigmented skin models, which attested that the proposed loaded EXT microemulsion produced an improved depigmenting effect compared to a commercial formulation. Hence, the developed microemulsion incorporating the cutite fruit extract is a promising novel formulation for treating hyperpigmentation conditions such as melasma.

## 5. Patents

GRATIERI, T. et al. Composição contendo extrato do fruto do Cutite (*Pouteria macrophylla*) (Lam.) Eyma em nanoemulsão e seu uso tópico despigmentante para tratamento de hipercromias. Depositante: Fundação Universidade de Brasília, Universidade Federal do Pará. BR 10 2021 008059 0. Depósito: 27 abr. 2021.

## Figures and Tables

**Figure 1 molecules-27-05982-f001:**
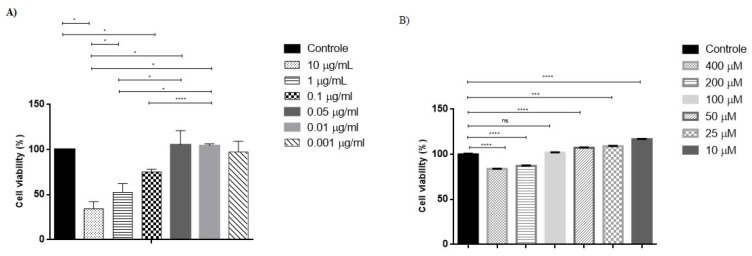
Effect of extract (**A**) and gallic acid (**B**) on cell viability. Melanocyte viability was assessed by MTT after 48 h incubation with cell culture media as positive control (C+), extract and gallic acid. Each bar represents the mean ± SD of 3 independent experiments. * *p* < 0.05; *** *p* < 0.001; **** *p* < 0.0001; ns = non-significant.

**Figure 2 molecules-27-05982-f002:**
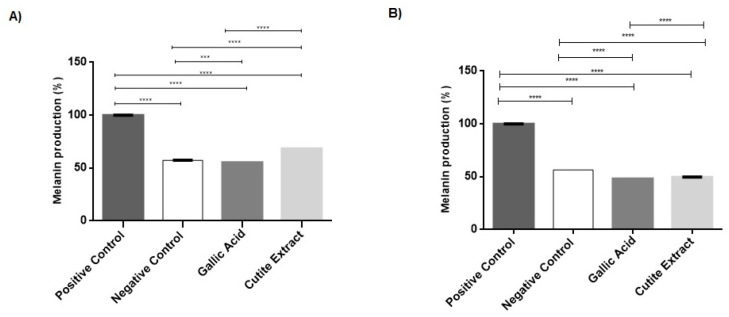
Melanocyte pigmentation was analyzed compared to IBMX-stimulated cells (positive control) and untreated cells (negative control). Intracellular melanin (**A**) production produced by B16F10 cells as well as melanin secretion in culture supernatant (**B**) after treatment with gallic acid and extract for 48 h, stored in an oven, were measured. Each bar represents an average of 3 determinations ± SD. *** *p* < 0.001; **** *p* < 0.0001.

**Figure 3 molecules-27-05982-f003:**
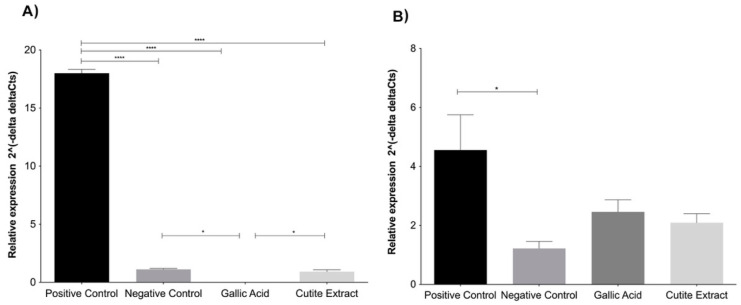
Levels of expression of MITF (**A**) and Tyrosinase (**B**) genes treated with gallic acid and extract. Each bar represents an average of 3 determinations ± SD. * *p* < 0.05; **** *p* < 0.0001.

**Figure 4 molecules-27-05982-f004:**
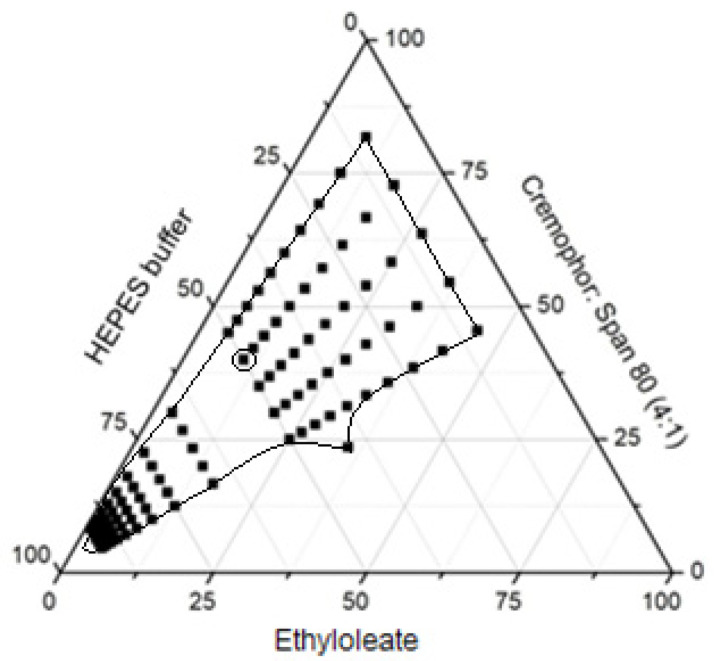
Pseudoternary diagram of formulation composed of oil (ethyl oleate), a mixture of surfactants (Cremophor^®^ and Span 80^®^ 4:1 *w*/*w*) dispersed in pH 4.5 HEPES buffer at room temperature. The dots correspond to the compositions that formed homogeneous, translucent systems classified as microemulsions.

**Figure 5 molecules-27-05982-f005:**
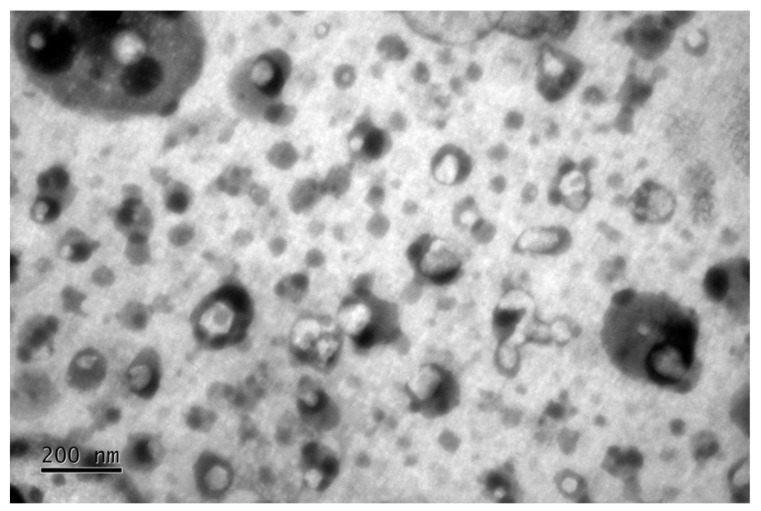
Image obtained by TEM of microemulsion with EXT corresponding to 0.05% GA incorporated.

**Figure 6 molecules-27-05982-f006:**
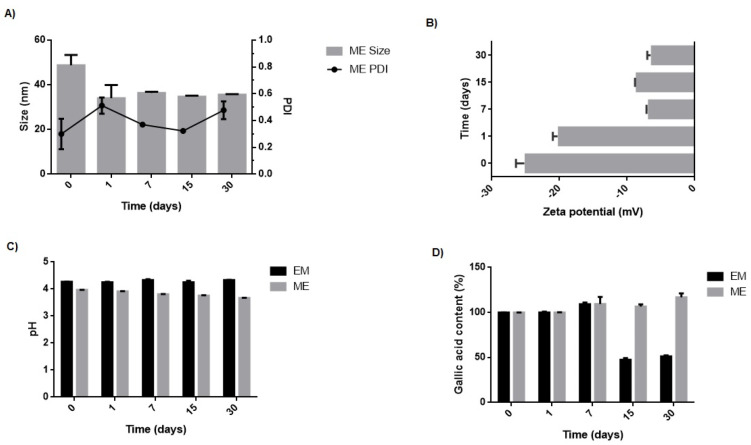
Evaluation of average size and PDI (**A**), zeta potential (**B**), pH (**C**), and GA content (**D**) of the microemulsion (ME) and conventional emulsion (EM) stored at room temperature over 0, 7, 15, and 30 days.

**Figure 7 molecules-27-05982-f007:**
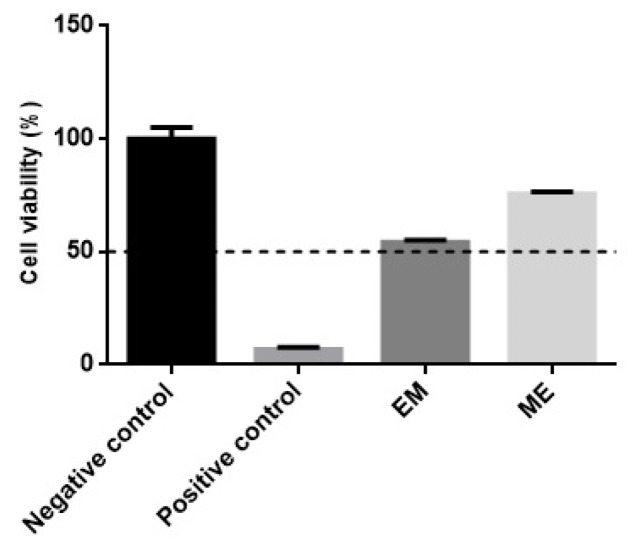
RHE irritability test of conventional emulsion (EM) and microemulsion (ME) compared to saline 0.9 % and sodium dodecyl sulfate solution at 5 % *m*/*v* (used as negative and positive controls, respectively). The dotted line delimits the minimum viability level for a potentially irritating sample. The data represent the means of three determinations ± standard deviation.

**Figure 8 molecules-27-05982-f008:**
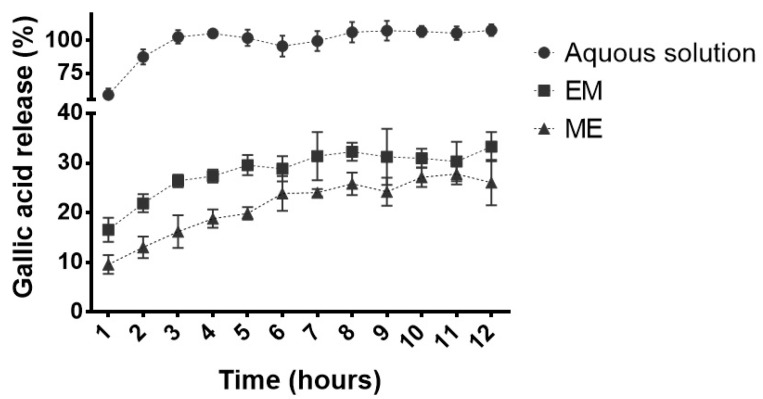
Gallic acid release profile from conventional emulsion (EM) and microemulsion (ME) formulations compared to an aqueous drug solution. All formulations contained 0.05% (*m*/*v*) of gallic acid (*n* = 5).

**Figure 9 molecules-27-05982-f009:**
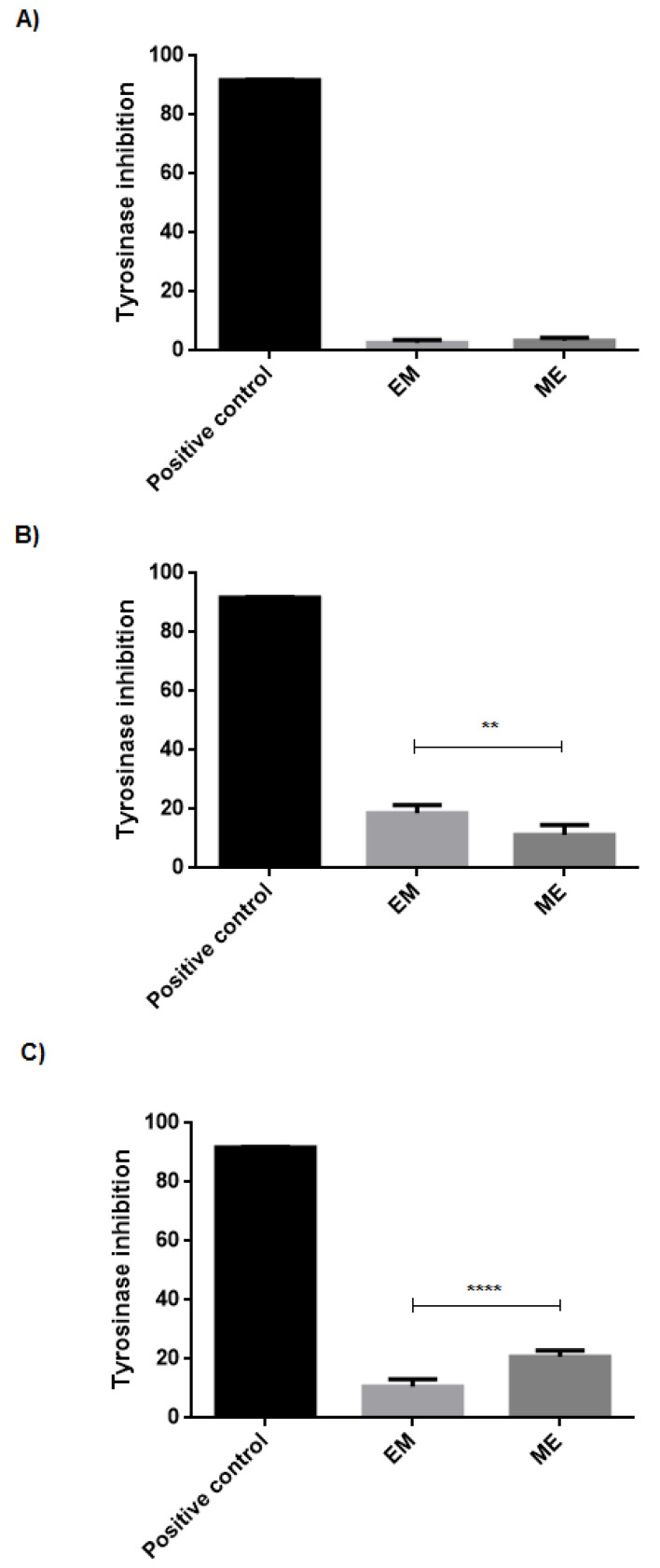
Tyrosinase inhibition by EXT contained in conventional emulsion (EM) and microemulsion ME at (**A**) receptor solution, (**B**) stratum corneum, and (**C**) remaining skin, after 6 h of in vitro skin permeation assay. Each bar represents an average of four determinations ± SD. ** *p* < 0.01; **** *p* < 0.0001.

**Figure 10 molecules-27-05982-f010:**
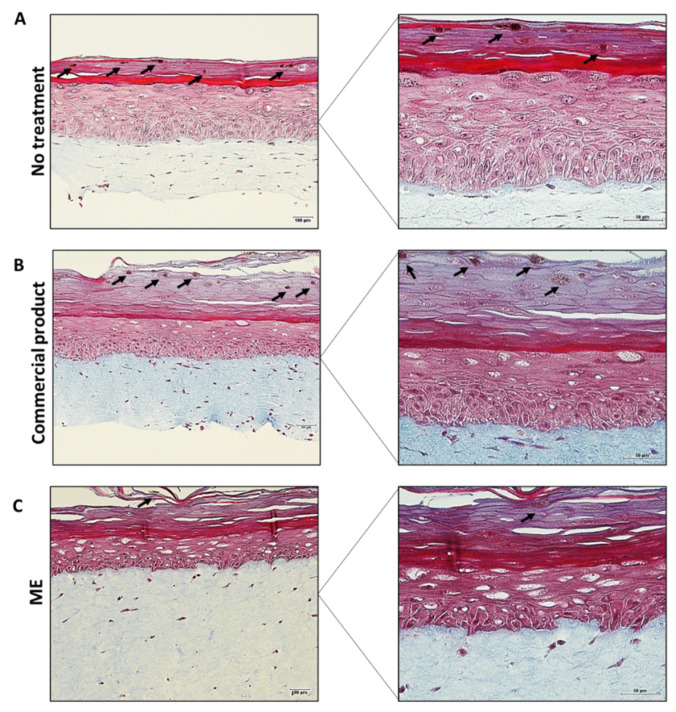
Histology of pigmented reconstructed human skin stained for Fontana Masson. (**A**) untreated skin; (**B**) skin treated with the commercial product; (**C**) skin treated with ME containing the extract. Left panel: Bar = 100 μm, original magnification 20x. Right panel: Bar = 50 μm, original magnification 40×. Arrows indicate the melanin spots. Data are representative of the three independent experiments.

## Data Availability

All data generated and analyzed during this study are available from the corresponding author on reasonable request.

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
