# Peer review of "Pouteria macrophylla* Fruit Extract Microemulsion for Cutaneous Depigmentation: Evaluation Using a 3D Pigmented Skin Model"

_molecules, 2022, doi:10.3390/molecules27185982_

Round 1
Reviewer 1 Report
1. I suggest adding in a 3D pigmented skin model to the title.
2. Were the collected fruits identified by an expert in botany? It would be interesting to include some photos of the fruit used for the development of this work, since not everyone knows it.
3. The study is interesting as well as the results obtained, however, it is necessary to demonstrate that the final product obtained is hypoallergenic in humans, since not all skins respond in the same way, regardless of whether a skin irritation test has been used. skin with reconstructed epidermis, so for this study to have strong support, a clinical study should be done.

Author Response
- I suggest adding in a 3D pigmented skin model to the title.
- We accepted the reviewer’s suggestion and added the skin model in the title as follows: “Pouteria macrophylla fruit extract microemulsion for cutaneous depigmentation: evaluation using a 3D pigmented skin model”.
- Were the collected fruits identified by an expert in botany? It would be interesting to include some photos of the fruit used for the development of this work, since not everyone knows it.
- Yes. The aerial parts and fruits were identified by comparison with authentic vouchers of Pouteria macrophylla (Lam.) Eyma exists in the Herbarium of Museum Emílio Goeldi (MG239766) by an expert in botany, employer of the Museum, city of Belém, state of Pará, Brazil. This information is now added to the manuscript (lines 331-333). Unfortunately, regarding the photos, we don’t have any good photos of our own (with publishing rights). There are several images in Google, but we don’t have a proprietary picture. For being a common fruit in the region, we didn’t think about that at the time and already processed all the fruits we had. For being a seasonal fruit, only after the next two months there will be fruits available again. We will collect them again for the next part of the study, a clinical trial, and we will keep that in mind for the next paper, thank you.
- The study is interesting as well as the results obtained, however, it is necessary to demonstrate that the final product obtained is hypoallergenic in humans, since not all skins respond in the same way, regardless of whether a skin irritation test has been used. skin with reconstructed epidermis, so for this study to have strong support, a clinical study should be done.
- Yes, we agree with the reviewer that the reconstructed epidermis provides evidence that the formulation is not irritating to the skin, but it lacks an immunogenic response. We do believe the formulation is hypoallergenic because all the excipients used are well-known in the cosmetic industry. Still, we are already planning to perform a clinical study, and the documents have already been submitted to the Local Ethics Committee for approval. The results of the clinical study will be submitted as a second paper. We believe the data presented in the paper already supports further studies and hence, is worth being published as soon as possible.
Reviewer 2 Report
The manuscript, entitled by Pouteria macrophylla fruit extract microemulsion for cutaneous depigmentation, mainly investigated the depigmenting ability of cutite fruit extract and its microemulsion formulation via melanocyte culture and 3D pigmented skin model. Results showed that cutite fruit extract had potent anti-melanogenic activity in vitro. The research approaches were appropriate, and the results sounded reasonable. Other comments were listed as the follow:
Specific remarks
# In the presenting study, authors hypothesized that not only GA but also other substances (ferulic acid and catechin) attributed to the anti-melanogenic ability. How about the content of other phytochemicals? Is there any data?
#Figure 3 and Figure 5 are low-resolution graphs and need to be improved.
# The part 3.5.2 Emulsion-gel said that the pH of the conventional formulation was 3.74, however, Figure 6C showed that it was above 4.
#The pH values of microemulsion gradually decreased and were lower than those of the conventional formulation during 30-day storage period, while the results of skin irritation test demonstrated that microemulsion presented higher cell viability. Was it the microemulsion without storage for irritation test? If so, how about the irritation result of microemulsion on day 30? Lower pH values may bring some adverse effects.
Minor Changes
#Line 84, the GA content should be expressed accurately, e.g. “13.4 mg/g DW”
#Line 347, it should be “section 3.2” ? Besides, it’s better that the section “Cutite fruit (Pouteria macrophylla) Lam. Emya extract obtention and characterization” changes to the position before the section “”LC-MS analyses.
#Line 356 (also Line 370), it should be 104 or 10^4 cells. Please change it.
Author Response
Reviewer 2#
The manuscript, entitled by Pouteria macrophylla fruit extract microemulsion for cutaneous depigmentation, mainly investigated the depigmenting ability of cutite fruit extract and its microemulsion formulation via melanocyte culture and 3D pigmented skin model. Results showed that cutite fruit extract had potent anti-melanogenic activity in vitro. The research approaches were appropriate, and the results sounded reasonable. Other comments were listed as the follow:
Specific remarks
# In the presenting study, authors hypothesized that not only GA but also other substances (ferulic acid and catechin) attributed to the anti-melanogenic ability. How about the content of other phytochemicals? Is there any data?
- Yes. In the introduction we cited some reference whose data were obtained in the same laboratory as ours in the UFPA, in Belém (lines 40-44). As follows: “Chemically, the cutite fruit is characterized by the presence of phenolic compounds, such as gallic acid (GA), p-coumaric, vanillic, ferulic, 3,4-dihydroxybenzoic, synaptic, caffeic, and flavonoids as quercetin and catechins [2–4]. Furthermore, the presence of GA as the main compound and biomarker, followed by other phenolic compounds, promotes its antioxidant action [2,5]”. For the extract used in this particular study we quantified the amount of GA and quercetin. The results are included in lines (83-87) as follows: “The EXT consists of several phenolic actives, with GA as a major phenolic component. GA content in the extract was calculated to be 13.4 mg/g, i.e., 1.34%. These values were over from those previously reported in the literature to fresh weight (4.72 mg/g) and very similar in relation to dry matter (12.5 mg/g) [2]. Other major component was quercetin (43.7 mg/g), which was in accordance with the literature [5].”.
#Figure 3 and Figure 5 are low-resolution graphs and need to be improved.
- Thank you for the observation. We replaced these pictures for one with better resolution.
# The part 3.5.2 Emulsion-gel said that the pH of the conventional formulation was 3.74, however, Figure 6C showed that it was above 4.
- We thank the reviewer for the observation. Phase 1 comprised 94.5% of pH 4.5 HEPES buffer. The pH of the formulation actually varied from 3.7 to 4.2. We consider this variation common for semisolid formulations and depends in fact of the pHmeter calibration. We corrected the text in line 433 as follows: “The pH of the formulation was 4.0± 0,3.”.
#The pH values of microemulsion gradually decreased and were lower than those of the conventional formulation during 30-day storage period, while the results of skin irritation test demonstrated that microemulsion presented higher cell viability. Was it the microemulsion without storage for irritation test? If so, how about the irritation result of microemulsion on day 30? Lower pH values may bring some adverse effects.
- We agree with the reviewer. Longer stability studies must be performed. But indeed, there was a minor pH oscillation within 30 days of storage, yet such oscillation was not even an statistical difference. The pH of the microemulsion oscillated from 4.0 ± 0.1 to 3.7 ± 0.1. So, we believe if this tendency persists in longer studies other buffers or preservatives should be added to the formulation. This was just a preliminary stability study and no preservatives were added. We added this to the discussion (lines 197-199). For the skin irritation tests formulations were prepared and stored at room temperature for one week previously the experiments (lines 445-447).
Minor Changes
#Line 84, the GA content should be expressed accurately, e.g. “13.4 mg/g DW”
- Corrected. Thank you.
#Line 347, it should be “section 3.2” ? Besides, it’s better that the section “Cutite fruit (Pouteria macrophylla) Lam. Emya extract obtention and characterization” changes to the position before the section “”LC-MS analyses.
- We accepted reviewers suggestion and changes the sections positioning.
#Line 356 (also Line 370), it should be 104 or 10^4 cells. Please change it.
- Corrected. Thank you.
Round 2
Reviewer 1 Report
I agree with the answers given by the authors, therefore, for me, the article can be accepted.